# Anti-Oxidation Agents to Prevent Dye Degradation in Organic-Based Host–Guest Systems Suitable for Luminescent Solar Concentrators

**DOI:** 10.3390/ma16020656

**Published:** 2023-01-10

**Authors:** Francesca Villafiorita-Monteleone, Mariacecilia Pasini, Chiara Botta

**Affiliations:** Istituto di Scienze e Tecnologie Chimiche “Giulio Natta” (SCITEC), Consiglio Nazionale delle Ricerche (CNR), Sede Via A. Corti 12, 20133 Milano, Italy

**Keywords:** photodegradation, photostability, luminescent solar concentrators, host–guest systems, Foerster Resonant Energy Transfer, organic dyes, green energy, luminescent materials, antioxidants

## Abstract

Luminescent solar concentrators (LSCs) have been extensively studied as they offer a practical solution to increase the efficiency of silicon-based photovoltaics (PVs). In this context, the use of natural and organic luminescent materials is desirable in order to obtain sustainable and environmentally friendly devices. Moreover, solution-processable organic host–guest systems based on Foerster Resonant Energy Transfer (FRET) processes offer the possibility to exploit a low-cost technique to obtain an efficient energy downshift from the UV–visible to red or deep red emissions in order to concentrate the radiation in the area of maximum efficiency of the PV device. Nevertheless, organic materials are subjected to photodegradation that reduces their optical properties when exposed to UV light and oxygen. In this work, we incorporated two different antioxidant molecules (i.e., octadecyl 3-(3,5-di-tert-butyl-4-hydroxyphenyl)propionate (Octa) and L-ascorbic acid (L-Asc)) in a three-dye host–guest system and studied the corresponding optical properties after prolonged irradiation times in air. It was found that the presence of the antioxidants, especially L-Asc, slowed the system’s photodegradation down whilst at the same time retaining high emission efficiencies and without interfering with the cascade Resonant Energy Transfer processes among the dyes inserted in the nanochannels of the host.

## 1. Introduction

As the human population grows, the development of sustainable, affordable and environmentally friendly energy production is crucial. Renewable energy, especially solar photovoltaic (PV) technology, provides the possibility for solving resource and environmental issues with the advantage of zero CO_2_ emissions [1,2,3]. Unfortunately, the cost of generating electricity from silicon solar cells remains high [4], not to mention that the waste generated during their production and after their end-of-life hurts the environment and cannot be ignored [5].

Luminescent solar concentrators (LSCs) are a widely studied and viable solution to the fabrication of sustainable and cheap PV systems characterized by high transparency, light weight, limited costs, synthetic versatility, high absorption coefficients and an ability to capture both incident and diffuse light [6,7,8]. Moreover, they can be fully integrated into architectural elements and become part of a building’s skin unit [9,10,11].

Basically, an LSC is a coating of luminescent materials consisting of dopants dispersed in an optically transparent host material (i.e., polymer matrices or glass substrates) acting as optical waveguide, in which the luminescent materials absorb both direct and indirect sunlight and then emit downshifted photons through a photoluminescence (PL) process (usually fluorescence) [6,12]. A large portion of the emitted light is concentrated via Total Internal Reflection (TIR) to the edges of the LSC, where a strip of PV cells may be attached, to collect the emitted light and convert it to an electrical current [13]. In this way, the possibility of concentrating the light radiation through Energy Transfer (ET) processes in the area of maximum efficiency of the device allows the minimizing of the size of the solar cells, with a consequent benefit in terms of production costs and environmental impact.

Up to now, various luminescent materials have been used as dopants, such as organic dyes [14,15,16,17], metal clusters [18,19], inorganic quantum dots [20,21,22,23,24], perovskites nanocrystals [25,26] and, more recently, carbon dots [27,28].

Despite the simple design and much needed applications, the widespread use of LSCs is still challenging, predominantly due to the photodegradation of both matrices and luminophores [6,29,30,31,32] and the reduced device efficiency, which in turn is linked to various light-loss processes [6,33,34,35].

In this context, potential approaches to minimize the absorption losses include the use of fluorescent dyes with a tunable Stokes shift [8,16,36,37], the use of luminophores with Aggregation-Induced Emission (AIE) properties [38,39] and, mostly, the use of organic or hybrid host–guest systems and the exploitation of Foerster Resonant Energy Transfer (FRET) processes, since they allow a significant improvement in the light-absorption efficiency while, at the same time, reducing reabsorption losses [40,41,42,43,44,45].

At the same time, to reduce the undesirable degradation effects of solar radiation on the luminophores, several approaches have been considered, ranging from the deliberate structural modification of dyes to the use of protective agents of various kinds. A variety of additives have been commercially developed, which can be used either before, during or after the incorporation of the dye in the device. Such additives may protect the dye, the matrix or, ideally, both [46,47].

In this work, we present a study on the photostability of deoxycholic acid (DCA)-based host–guest materials containing three linear dyes (Figure 1a) (diphenylhexatriene (DPH) emitting in the blue (Figure 1b), 4,7-di(2-thienyl)-benzo[1–3]thiadiazole (DBT) emitting in the green (Figure 1c), and 4,7-bis(5′-hexyl-2,2′-bithienyl-5-yl)-2,1,3-benzothiadiazole (DBTT) emitting in the red (Figure 1d)) for the development of photostable LSCs.

These systems have previously been demonstrated to be able to shift the near UV and visible light to the deep red with a very high efficiency, thanks to controlled FRET processes among the dyes inserted in the nanochannels characteristic of a DCA organic host [42,48]. Here, we exposed thin films of these materials to prolonged irradiation in order to test their performances in LSC devices.

To avoid the photodegradation of the systems, we added some antioxidant molecules and studied the effect of their presence on the materials’ optical properties.

## 2. Materials and Methods

### 2.1. Materials

Deoxycholic acid (DCA), 1,6-diphenyl-1,3,5-hexatriene (DPH), octadecyl 3-(3,5-di-tert-butyl-4-hydroxyphenyl)propionate (Octa) and L-ascorbic acid (L-Asc) were purchased from Sigma Aldrich and used without further purification. 4,7-Di(2-thienyl)-benzo[1–3]thiadiazole (DBT) and 4,7-bis-(5′-hexyl-2,2′-bithienyl-5-yl)-2,1,3-benzothiadiazole (DBTT) were synthesized as reported elsewhere [49,50,51]. All solvents were of analytical grade.

The DCA mother solution was prepared with a concentration of about 10^−2^ M, and antioxidant and dye mother solutions were prepared with a concentration of about 7 × 10^−3^ M in THF. Reference samples were prepared by mixing DCA and dye mother solutions in order to prepare co-inclusions with the following ratios: DCA:DPH=3:1 and DPH:DBT:DBTT = 96.9:2.9:0.2. Antioxidant mother solutions with a concentration of about 2 × 10^−3^ M in THF were mixed with co-inclusion mother solutions in order to have DBT:antioxidant concentrations of 90 and 10, 80 and 20 and 95 and 5 wt.%.

Films of DCA inclusion compounds with and without antioxidants were obtained by spin-coating the corresponding solutions at 500 rpm for 60 s on 2 × 2 cm^2^ quartz substrates. The resulting films had a thickness of about 150–200 nm.

### 2.2. Optical Characterization

PL measurements on solid state samples in air at room temperature were obtained with a SPEX 270 M monochromator equipped with a N_2_-cooled charge-coupled device excited with a monochromated Xe lamp. The spectra were corrected for the instrument response. PL quantum yields (QYs) (The Photoluminescence Quantum Yield of a molecule or material is defined as the number of photons emitted as a fraction of the number of photons absorbed [52]) of solid state materials were obtained by using a homemade integrating sphere and by correcting the spectra for the background of the exciting lamp according to [48].

For accelerated photodegradation experiments, all samples were subjected to continuous illumination in air by means of a UV-lamp (Hamamatsu L9566 series) with a 300–400 nm band-pass excitation filter at a 30 mm distance, where the lamp intensity was about 410 mW/cm^2^ (Figure 1).

Fluorescence spectra were collected after each period of exposure (0, 5, 10, 30, 45, 60, 80, 90, 105, 120, 180, 300, 480, 960, 1800, 3600 s) with the above-mentioned apparatus. The photodegradation was followed by monitoring the decrease in the overall emission and/or in each dye emission band (i.e., in the 400–475 nm range for DPH, in the 475–585 nm range for DBT and in the 585–750 nm range for DBTT). The decrease in the emission intensity for both the DCA-DPH:DBT and the DCA-DPH:DBT:DBTT samples was analyzed, with a bi-exponential fit with and without antioxidants.

## 3. Results and Discussion

As previously reported, thin films of organic host–guest systems based on the use of DCA, obtained from bovine bile, as the host involve the use of sustainable and natural components and can be obtained with low-cost self-assembly technology [42,48].

These materials can be used to obtain films with a three-dye system able to efficiently downshift the near-UV-visible light to deep red thanks to controlled FRET processes, in order to develop LSCs with reduced energy consumption [42]. Moreover, to obtain such highly efficient host–guest systems, it is sufficient to use a small amount of DBT with respect to DPH to shift the system’s emissions from blue to green, and thus to use an even smaller quantity of DBTT with respect to DBT to shift the emission from green to red, as demonstrated elsewhere [42].

As shown in Figure 1b–d, the three dyes were characterized by optical properties ideal for a cascade FRET process to shift the energy from the near-UV (DPH absorption) region to the green (DBT emission) region and, finally, to the deep red (DBTT emission) region. This system worked because the absorption of each acceptor overlapped with the emission of each donor, and the relative Foerster radii led to an efficient funneling from the UV region to the deep red region through a multi-step FRET process (DPH → DPH, DPH → DBT, DPH → DBTT and DBT → DBTT) [42].

Indeed, the thin films of co-inclusions of DPH, DBT and DBTT in DCA showed emissions from all chromophores when the DPH was excited (Figure 2). It was, in fact, possible to recognize the emission bands of each dye: DPH presented an emission peak in the 400–475 nm range, DBT in the 475–585 nm range and DBTT in the 585–750 nm range.

One major drawback against the wide diffusion of organic systems such as the one just described is the photostability of the materials that unfortunately are subjected to degradation when exposed to UV light and oxygen. The inclusion of organic dyes into a DCA host is expected to protect the dyes from photodegradation, since the host reduces their exposure to oxygen [42,48]. Thus, to assess their use in LSC devices, we performed an accelerated aging test on host–guest thin films by exposing them to prolonged irradiation in air. Aging the films with a UV lamp in air induced an emission quenching, as shown in Figure 3**,** where the emission spectra of the bare DCA-DPH:DBT:DBTT films are compared for increasing irradiation times.

It is possible to notice an evident quenching in the overall fluorescence intensity with increasing light exposure times, even though some differences can be highlighted. Specifically, a sharp emission decrease was observed for the peaks associated with DBT and DBTT (in the 475–585 nm and 585–750 nm ranges, respectively (Figure 2)) in the host–guest system, whereas a slower and less pronounced reduction in the PL intensity was detected for the peak associated with DPH (in the 400–475 nm range, Figure 2). This behavior is better highlighted in Figure 4, where the changes in the emission intensities are reported as the ratio between the emission intensity measured after a specific time interval (*I*) and the initial emission intensity (*I*_0_).

These different quenching behaviors could be due to the differences in the chemical structures of the chromophores. In fact, thiophene-based materials are more easily subjected to photooxidation, as previously reported [53,54,55,56]. Indeed, the presence of thiophene groups in both DBT and DBTT leads to a larger number of sites ideal for oxidative reactions.

To assess the role of the antioxidants on the overall stability, we started with DCA-DPH:DBT co-inclusion compounds and evaluated the effect of different additive concentrations (5 wt.%, 10 wt.% and 20 wt.% with respect to the DBT concentration). We chose two different antioxidant molecules: octadecyl 3-(3,5-di-tert-butyl-4-hydroxyphenyl)propionate (Octa) and L-ascorbic acid (L-Asc) (Figure 2). The former is a synthetic antioxidant, largely used in plastics and food packaging [57] and is large enough to act as stopcock by inserting its long alkyl chain inside the host channel and blocking the entrance thanks to the bulky benzene derivative [58,59]. The latter is the L-enantiomer of ascorbic acid, a natural water-soluble vitamin, small enough to enter the DCA nano-channels and suitable for green chemistry [60].

We exposed the host–guest systems in air to UV light for increasing time intervals and monitored the photodegradation of the organic components by means of fluorescence spectroscopy.

As shown in Figure 5, after different irradiation times, almost all the additive concentrations led to an improvement in the resistance to photodegradation when compared with the bare sample. According to the experiment, the best results were achieved with a concentration of 10 wt.% for both antioxidants: higher antioxidants concentrations had a negative or no effect because of the changes in the FRET process.

The comparison of the decay times (Table 1), obtained as averages of the bi-exponential fits, confirmed the choice of using 10 wt.% of additive as the best-performing anti-oxidant concentration.

Thus, for the complete systems characterized by the co-inclusion of the three dyes, we considered the addition of 10 wt.% of L-Asc or Octa. Figure 6 shows the degradation of all the samples’ emissions as a function of different UV light irradiation times. In addition, for the complete system, the presence of L-Asc seemed to reduce the photodegradation with respect to both the bare system and the sample with Octa. In fact, after 1h of irradiation in accelerated photodegradation conditions, the intensity was still 85%. Moreover, as reported in Table 2, the lifetime increased from 5.9 h to 12.1 h with the addition of 10 wt.% L-Asc to the DCA matrix. This result might be due to the fact that L-Asc is a small molecule that penetrates the DCA nanochannels and performs its antioxidant function within close proximity of the dyes [60].

To further assess the role of the antioxidant molecules added to the co-inclusion compounds as means for improving the photostability, we compared the additive-containing systems to bare co-inclusions prepared in an oxygen-free environment and then subjected them to UV irradiation in air. Figure 7 shows the PL emission intensity ratios of the DCA-DPH:DBT:DBTT samples prepared in air or in an inert atmosphere and of the systems containing L-Asc or Octa prepared in air.

As shown in the following plot and in Table 2, the presence of L-Asc led to an initial lower photodegradation when compared to the bare samples prepared in the absence of oxygen, showing an emission loss of a maximum of 25% despite being prepared in air. As stated above, this might be due to the small dimensions of this particular antioxidant that easily penetrates inside the DCA nanochannels and performs its role in eliminating reactive oxygen species directly in situ [60].

Finally, we performed a last aging test, exposing all the samples to air and natural light for one week and measuring their absolute PL QYs before and after UV exposure. Table 3 shows the values for the bare samples prepared either in the absence of oxygen or in air and for the additive-containing systems prepared in air, before and after the aging experiments. Once again, the samples prepared with 10 wt.% of L-Asc showed the best results, with no emission losses whatsoever. This result demonstrates that the inclusion of L-Asc as an anti-oxidation agent in DCA films containing three organic dyes reduced the emission-quenching processes typical of organic materials, allowing the preparation of thin films from solutions in air with simple and low-cost techniques.

## 4. Conclusions

We studied the possibility of increasing the photostability of organic host–guest systems suitable for LSC applications by comparing two different antioxidant molecules: octadecyl 3-(3,5-di-tert-butyl-4-hydroxyphenyl)propionate (Octa), a synthetic antioxidant, largely used in plastics and food, and L-ascorbic acid (L-Asc), a natural water-soluble vitamin found in fruit and vegetables. In particular, the use of L-Asc enables the obtaining of natural and environmentally friendly compounds and the improvement of the system’s photostability without recurring to encapsulation or preparation procedures involving an inert atmosphere, thus following a very simple and low-cost technique (i.e., deposition in air of solutions containing the different compounds) for the production of organic thin films compatible with the realization of low-cost LSCs.

Although this work is in its infancy, these values and the stability of the presented systems as optically active materials are a promising starting point for the development of sustainable and stable LSCs.

## Data Availability

The data presented in this study are available within the article and in ref. [42].

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
