# Peer review of "Anti-Oxidation Agents to Prevent Dye Degradation in Organic-Based Host–Guest Systems Suitable for Luminescent Solar Concentrators"

_materials, 2023, doi:10.3390/ma16020656_

Round 1

Reviewer 1 Report

See the attached document.

Author Response

We thank the reviewer for all the proper comments that allowed us to improve the manuscript. In the attached file the point by point answer.

Reviewer 2 Report

In the manuscript entitled "Bio-Based Matrix to Prevent Dye Degradation in Luminescent 2 Solar Concentrators", the authors have incorporated two different antioxidant molecules (octadecyl 3-(3,5-di-tert-butyl-4- hydroxyphenyl)propionate (Octa) and L-ascorbic acid (L-Asc)) in a three-dye host-guest system and studied the corresponding optical properties after prolonged irradiation times in the air. The manuscript is well written and can be further improved for possible publication if the following recommendations are incorporated. 

1) In my opinion the word Bio-based matrix is not appropriate, the word organic is enough.

2) In Figure 1 (b,c,d) , the same colors are used for normalized PL and normalized optical absorption (without any marker) for each organic. Although from the wavelength associated with peak values we get a little idea. But it is better authors must differential both curves in each figure. 

3) Line 99. authors give spin coating rate and time duration but do not give the thickness of each film.  

4) In Figure 7 authors show that the inclusion of L-Asc  as anti-oxidation agent in DCA films containing three organic dyes reduces the emission-quenching processes typical of organic materials. It will be better if some technical reasons should present to justify your results.

5) Similarly in Figure 5, authors reported that the best results were achieved with a concentration of 10 wt% for both antioxidants: higher antioxidant concentrations have a negative effect probably because of the change in the film morphology and in the FRET process. But it will be better if authors justify their results with SEM characterization. 

Author Response

We thank the reviewer for all the proper comments that allowed us to improve the manuscript. In the attached file the point by point answer

Round 2

Reviewer 1 Report

The authors have made an effort to answer my suggestions, mainly by including a temporal analysis  of the photodegradation (fittings of the curves). However, in my opinion, this analysis should be included in the discussion of the results instead of in the optical characterization section, as has been done. In addition, errors in the results of the fittings are missing.

Author Response

We thank Reviewer 1 for the further request to better evidence the fitting results, and to include the fit errors, that we forgot to insert in the old Table 1. In the second revision we moved Table 1 from the optical characterization section to the Results and Discussion section, where two new tables (Table 1 and  Table 2) include the parameters used for the fits and the corresponding standard deviation errors and decay times. Moreover, we add further comments to evidence these results, as the comment to Figure 6 ”after 1h irradiation in accelerated photodegradation conditions, the intensity is still 85%. Moreover, as reported in Table 2, the lifetime increases from 5.9 h to 12.1 h with the addition of 10 wt% L-Asc to the DCA matrix." We modified the text from line 123 to line 125, from line 214 to line 218, and from line 223 to line 226. Accordingly, we changed the captions of figures 5-7 and the previous Table 1 has become Table 3.

Reviewer 2 Report

As authors have incorporated all my suggestions. Therefore, I recommend the manuscript for possible publication. 

Author Response

We thank Reviewer 2 for the positive evaluation of the first revision